# Simple and Convenient Method for Assessing the Severity of Bleeding during Endoscopic Prostate Surgery and the Relationships between Its Corresponding Surgical Outcomes

**DOI:** 10.3390/diagnostics13040592

**Published:** 2023-02-06

**Authors:** Shu-Chuan Weng, Shu-Han Tsao, Han-Yu Tsai, Horng-Heng Juang, Yu-Hsiang Lin, Phei-Lang Chang, Chien-Lun Chen, Chen-Pang Hou

**Affiliations:** 1Department of Health and Management, Yuanpei University of Medical Technology, Hsinchu 330, Taiwan; 2Bachelor Degree Program of Senior Health and Management, Yuanpei University of Medical Technology, Hsinchu 330, Taiwan; 3Department of Urology, Chang Gung Memorial Hospital at Linkou, Taoyuan 333, Taiwan; 4School of Medicine, Chang Gung University, Taoyuan 333, Taiwan; 5Department of Anatomy, School of Medicine, Chang Gung University, Kwei-shan, Taoyuan 333, Taiwan; 6Graduate Institute of Clinical Medical Sciences, College of Medicine, Chang Gung University, Taoyuan 333, Taiwan

**Keywords:** prostate, endoscopic, thulium laser, enucleation, bleeding, hematuria

## Abstract

Bleeding during endoscopic prostate surgery is often overlooked, and appropriate measurement techniques are rarely applied. We proposed a simple and convenient method for assessing the severity of bleeding during endoscopic prostate surgery. We determined the factors affecting bleeding severity and whether they affected the surgical results and functional outcomes. Records from March 2019 to April 2022 were obtained for selected patients who underwent endoscopic prostate enucleation through either 120-W Vela XL Thulium:YAG laser or bipolar plasma enucleation of the prostate. The bleeding index was measured using the following equation: irrigant hemoglobin (Hb) concentration (g/dL) × irrigation fluid volume (mL)/preoperative blood Hb concentration (g/dL) × enucleated tissue (g). Our research revealed that patients who underwent surgery employing the thulium laser, those aged over 80 years, and those with a preoperative maximal flow rate (Qmax) of more than 10 cc/s experienced less surgical bleeding. The patients’ treatment outcomes differed depending on the severity of the bleeding. Enucleating prostate tissue was easier in the patients with less severe bleeding, who also had a lower risk of developing urinary tract infections and an improved Qmax.

## 1. Introduction

Benign prostatic hyperplasia (BPH) is the most common cause of lower urinary tract symptoms (LUTSs) in men [1]. Aging has a marked impact on histological BPH, increasing the risk of developing BPH to 90% in men aged between 81 and 90 years [2]. The prevalence of symptomatic BPH increases with age, from 44% in men aged 40 to 59 years to 70% in men aged over 80 years [3,4]. BPH is caused by the growth of epithelial and stromal cells inside the transition zone of the prostate gland, inducing obstructive and storage symptoms during voiding [5]. The medical and surgical management of BPH with LUTSs is a growing concern because the burden of BPH-related LUTSs is increasing more quickly than that of any other urological illness [6]. Although pharmacological medication (α blockers) is the first-line treatment for BPH/LUTSs, endoscopic surgery is a suitable option for those with moderate-to-severe LUTSs and for patients with BPH-related complications [7]. In addition to transurethral resection of the prostate (TURP), which has been regarded as the gold-standard treatment, a variety of laser systems and techniques have been introduced to reduce surgical blood loss, provide a clearer vision of the surgical field, shorten catheterization time, lower surgical morbidity, and improve quality of life (QoL) [8,9]. Endoscopic prostate enucleation provides favorable outcomes for BPH with benign prostate obstruction (BPO) [10]. 

A common complication of all methods of endoscopic prostatectomy is intraoperative bleeding, which results in a rate of blood loss necessitating transfusion of 0.4% to 7.1% [11,12]. Recently, the transfusion rate has declined because of advancements in resectoscopes, optics, anesthetics, and energy sources. However, moderate intraoperative bleeding occurs frequently and might have a negative impact on surgical results [13]. Intraoperative bleeding impairs surgical vision, which increases operating time, increases the risk of fluid overabsorption, increases the need for irrigation fluids, and ultimately increases the risk of surgical complications [13]. Massive perioperative bleeding that impairs visibility in the surgical area may also increase the difficulty of identifying anatomical landmarks. Previously, estimations of blood loss were made using indicator dilution methods [14,15] or visual analog scales [16] or were based on changes in the hemoglobin (Hb) level in the blood drawn postoperatively [17,18]. The use of artificial intelligence to interpret surgical videos to assess the severity of blood loss has also been reported [19]. None of these methods are objective, accurate, or economical. In addition, whether the severity of blood loss at the time of endoscopic prostate surgery affects short-term, medium-term, and long-term prognosis after the procedure has rarely been discussed in the literature. In this study, we proposed a simple and convenient method for assessing the severity of bleeding during prostate endoscopic surgery, namely the bleeding index (BI). We also explored factors affecting the severity of blood loss and whether blood loss affects surgical results, especially functional outcomes. 

## 2. Materials and Methods

### 2.1. Patient Selection and Evaluation

Records for the period of March 2019 to April 2022 were obtained for selected patients with symptomatic BPH who underwent 120-W Vela XL thulium:YAG laser prostate enucleation or bipolar plasma enucleation of the prostate (B-TUEP) in the geriatric urology department of Chang-Gung Memorial hospital, Taiwan, following institutional review board approval (IRB number: 201900094B0C502). A single experienced surgeon conducted all the procedures. The patients completed consent forms and were free to select their own treatment plan. Before surgery, every patient underwent a thorough evaluation including medical history taking, physical examination, digital rectal examination (DRE), serum prostate-specific antigen (PSA) assessment, and transrectal ultrasound of the prostate (TRUS). The patients on anticoagulant or antiplatelet medication all discontinued the medication as directed. To exclude the possibility of prostate cancer, each patient with a PSA level higher than 4 ng/mL received a TRUS biopsy if an abnormality was discovered during their DRE or TRUS. The following measurements were taken to determine each patient’s ability to void: voiding volume (VV), postvoid residual urine volume (PVR), preoperative maximal flow rate (Qmax), International Prostate Symptom Score (IPSS), and IPSS-QoL score. The following criteria were required for patient inclusion: prostate volume ≥ 30 cm^3^, IPSS ≥ 20, Qmax ≤ 15 mL/s, and Eastern Cooperative Oncology Group performance status ≤ 1 [20]. All patients underwent medical treatment for at least 3 months before surgery and met the surgical requirements for BPO [21]. Patients were excluded if they had a history of prostate surgery or active malignant disease. Patients whose postoperative pathology report revealed prostate cancer were also excluded. Also excluded were patients who reported neurogenic bladder or LUTSs ascribed to reasons other than BPH. During postoperative evaluation, if the clinician re-prescribed drugs that would interfere with urination, such patients were not included in the next follow-up.

### 2.2. Surgical Equipment and Techniques

The patients underwent B-TUEP procedures that employed an Olympus SurgMaster Electrosurgical Unit UES-40 bipolar generator and Olympus OES Pro Resectoscope (Olympus Europa, Hamburg, Germany). Cutting and coagulation were performed using normal energy values of 200 and 120 W, respectively. The enucleation and resection energies were 60 and 120 W, respectively. The surgical technique followed the procedure described by Liu et al. [22]. The 120-W Vela XL Thulium:YAG laser (Boston Scientific, Marlborough, MA, USA) with a continuous wavelength of 1.94 μm was used for all laser enucleation procedures. The fiber was introduced using the Olympus 26F continuous flow resectoscope (Olympus Europa, Hamburg, Germany). All procedures involved irrigation using a 0.9% sodium chloride solution. The prostate tissue that had been enucleated was ground and evacuated from the bladder by using a Wolf Piranha morcellator (Richard Wolf GmbH, Knittlingen, Germany) [23]. 

### 2.3. BI Calculation

The BI during the enucleation procedure was measured using the HemoCue Plasma/Low Hb photometer Ängelholm, Sweden) [24,25] and the following equation: Irrigant Hb concentration (g/dL) × irrigation fluid volume (mL)/preoperative blood Hb concentration (g/dL) × enucleated tissue (g).

The patients had blood drawn 1 day before the procedure, and the blood Hb concentration was measured. The Hb concentration was obtained from the irrigants collected during the procedure. To avoid blood coagulation, 15,000 IU of heparin was added to every 10 L container of collected irrigants during the operation [26]. 

### 2.4. Postoperative Care

At the end of each procedure, a three-way Foley catheter (22 Fr) was placed into the bladder to enable continuous irrigation. The prostate was not compressed with a catheter balloon to achieve hemostasis. Unless unanticipated adverse events necessitating delaying catheter removal occurred, catheters were removed on postoperative day 2. Antibiotics were administered both preoperatively and postoperatively following the recommended protocol [27]. When a patient exhibited signs of a postoperative infection, appropriate antibiotics were administered in accordance with bacterial culture and drug sensitivity studies. Regardless of whether urological drugs were administered prior to surgery, all patients received 0.4 mg of tamsulosin once a day for 1 week postoperatively. 

### 2.5. Follow-Up and Outcome Evaluation

Prostate tissue removed after surgery was sent for pathological examination. The enucleated tissue percentage is the percentage of the quotient obtained according to the volume of prostate tissue that was surgically removed and the volume of the prostate transition zone. After the patients were discharged from the hospital, they returned for follow-up visits at 2, 3, and 6 months. During the visits, the IPSS, IPSS-QoL score, Qmax, VV, and PVR were evaluated, and complications were recorded. If a patient had urinary tract infection (UTI) symptoms and the clinician prescribed antibiotics, we defined this as a UTI episode. All drugs related to urination were discontinued to ensure the objectivity of the effect of the procedure. However, if the clinician judged the patient to be in need of medicine, the patient was not included in the subsequent follow-up(s) after the current assessment was completed and recorded. 

### 2.6. Statistics

The chi-square test, independent-sample *t* test, and Pearson correlation (r) were used to analyze the relationship between the BI and the variables of the patients. The patients were divided into three groups based on the BI by using a quartile method ordered as follows: low: <Q1, medium: Q1–Q3, and high: >Q3. Changes in urodynamic parameters between each observation time point were analyzed using a one-way analysis of variance. The significance level for all statistical analyses was *p* < 0.05. All data were analyzed using the statistical software IBM SPSS (version 25, IBM: Armonk, NY, USA).

## 3. Results

Data from 226 patients who underwent endoscopic prostate enucleation were recorded and examined. The postoperative pathology reports of 11 patients revealed that they had prostate cancer, and these patients were therefore excluded. A total of 215 patients met the inclusion criteria. None of the patients required blood transfusion postoperatively. During the follow-ups, six patients dropped out of the study, and 49 took medication for symptom relief that might have affected their urination. Finally, a total of 150 participants completed the 6-month observation period. The baseline data (preoperative and perioperative) of the patients are summarized in Table 1. Among the variables, only the enucleated tissue percentage had a negative linear correlation with the BI (*r* = −0.235, *p* < 0.001); that is, the more severe the surgical bleeding was, the lower the amount of tissue that could be enucleated during enucleation surgery was. Other factors included age, prostate volume, preoperative PSA level, IPSS, urodynamic studies, operation time, and length of hospital stay, all of which had no linear correlation with the BI. The relationship between categorical variables and the BI is reported in Table 2. The patients who underwent prostate enucleation performed using a thulium laser had a lower BI than did those who underwent prostate enucleation performed using a conventional bipolar resection loop (5.73 vs. 12.20, *p* < 0.001). The patients who were aged older than 80 years had a lower BI than did those who were aged younger than 80 years (5.59 vs. 8.36, *p* = 0.019). The patients with a Qmax ≥ 10 cc/s had a lower BI than did those with a Qmax < 10 cc/s (6.08 vs. 9.08, *p* = 0.008). Other variables, including having a catheter at hospital admission, undergoing surgery for urinary retention, and having a prostate greater than 80 mL in size, were nonsignificantly related to the BI. In addition, none of the patients’ comorbidities, including diabetes mellitus, hypertension, coronary artery disease, congestive heart failure, arrhythmia, stroke, and renal insufficiency, were significantly related to BI. 

We used the quartile method to divide the patients into three groups on the basis of their BI values from low to high as follows: low: <Q1, medium: Q1–Q3, and high: >Q3. As illustrated in Figure 1, our data revealed that the patients in the low-BI group had the highest prostate removal percentage, significantly higher than that of the medium- and high-BI groups (117% vs. 80% and 72%, *p* < 0.001). Table 3 presents the relationships between various surgical bleeding severities and the postoperative outcomes. Twenty-six of the patients returned to the emergency department within 1 month after the operation, most for treatments for the following mild complications (Clavien–Dindo classification grade I and II): hematuria (ten), UR (eight), orchitis (one), and acute pyelonephritis (one); the other six patients returned to the emergency department for reasons not related to the surgery. Our statistics indicated no significant difference between the groups in terms of postoperative hospital stay, incidence of urethral strictures, postoperative UR, or rate of returning to the emergency room. However, bleeding severity was associated with the development of postoperative UTIs. Our study revealed that the proportion of postoperative UTIs was significantly lower in the patients with a low BI than in the other two groups (18.8% vs. 38.8% and 31.4%, *X*^2^ = 6.07, *p* = 0.046).

The patients’ functional outcomes after surgery over time are presented in Figure 2, Figure 3, Figure 4 and Figure 5. Figure 2 depicts the change in IPSSs over time after the procedure. Our study revealed that the improvement in IPSSs in the third and sixth month postoperatively was more significant than that in the second week postoperatively; those in the third and sixth months were the same. At these three observation time points, no significant difference in IPSS change was observed among the groups with different BIs. Similarly, at these observation time points, we noted no significant difference in the extent of Qmax improvement among the groups with different BIs. Although the graph reflects a trend in which a smaller BI indicates greater improvement in the Qmax, the difference between them was nonsignificant. Figure 4 illustrates the changes of VV at different observation time points. The improvement in VV was significantly larger at 6 months postoperatively than in the second week and third month postoperatively. However, we observed no significant difference in VV improvement among the groups with different BIs. Our patients exhibited a marked reduction in PVR postoperatively, as illustrated in Figure 5, with the reduction 6 months after the operation being significantly larger than that at 2 weeks postoperatively. However, no significant difference was observed in PVR improvement among the groups with different BIs.

## 4. Discussion

Bleeding during endoscopic prostate surgery is often overlooked, and appropriate measurement techniques are rarely applied. The lowering of blood Hb is widely employed as a measure of blood loss in the literature [17,18,28]. However, this method is inaccurate because it neglects the physiological function of the autoregulation of Hb. Because of the simultaneous loss of red cells and plasma after acute hemorrhage, Hb and hematocrit levels may remain normal, which becomes apparent after the patient’s plasma volume is restored either naturally or by using intravenous fluids [29]. Blood loss and hematopoiesis occur simultaneously in normal physiological conditions. During bleeding, hematopoietic stem cells divide and lead to more committed progenitors, producing all lineages of blood cells, including erythrocytes [30]. The speed of hematopoiesis in different patients varies by age, health status, and comorbidity. Therefore, objectively evaluating the severity of surgical bleeding by directly measuring Hb in the blood is difficult, especially in endoscopic prostate surgery where the blood loss is relatively small. One study reported a negligible linear relationship between measured blood loss and the relative and absolute change in blood Hb from before TURP to the morning after the surgery (*r*^2^ = 12–13%) [26]. Fagerström et al. proposed a simple and effective method for measuring blood loss in prostate surgery [26] and applied this method to verify that bipolar TURP causes less bleeding than does the monopolar technique. Our method is a modification of these researchers’ method. Our proposed BI reflects the surgical blood loss per cubic centimeter of prostate enucleation. Because the amount of blood loss in prostate surgery is related to the resection weight [31,32], calculating the amount of blood loss corresponding to each unit of the enucleated prostate can objectively reflect blood loss severity.

The first goal of our study was to identify factors affecting the severity of bleeding. None of the continuous variables had a linear correlation with the BI. However, further analysis of the categorical variables revealed that the severity of blood loss was significantly lower in the patients undergoing prostate enucleation performed using a thulium laser than in those undergoing prostate enucleation performed using a bipolar resection loop. These findings are comparable to those in related studies [32,33,34,35,36], although different methods for measuring blood loss were applied. Our study verified the advantage of thulium laser for preventing bleeding during prostate surgery. Another finding of our study is that patients aged older than 80 years have a higher BI than their younger counterparts do. This may be related to the relative insufficiency of pelvic blood flow in older adult patients. According to Berger et al., prostate blood supply becomes impaired with age, which may contribute to adenoma formation [37]. This vascular damage can also lead to chronic ischemia, which may contribute to the pathogenesis of BPH in aging men [38]. Sugaya et al. revealed that the average common iliac vein blood flow velocity was significantly lower in men with chronic prostatitis and overactive bladder [39], indicating that pelvic blood congestion is responsible for LUTSs. Our finding that the BI is relatively higher among patients with a lower Qmax (<10 cc/s) also supports this view. We speculated that venous blood stasis in the prostate of these patients resulted in increased blood loss during enucleation. We also analyzed factors potentially affecting the severity of surgical bleeding, including the patients’ comorbidities. However, our results indicate that the comorbidities were nonsignificantly associated with the severity of bleeding during prostate enucleation.

The second goal of our study was to determine whether the severity of bleeding during prostate enucleation impacts the surgical outcomes. We hypothesized that the more severe the bleeding was during the operation, the higher the rate of complications and the less favorable the functional outcome would be. Massive intraoperative bleeding can impair surgical area visibility and increase the difficulty of identifying surgical landmarks between the adenoma and the transition zone prostatic capsule. Contrarily, reduced blood loss during the enucleation procedure alleviates this problem and reduces the stress on the surgeon induced by profuse bleeding. Our results indicate that the severity of bleeding during surgery had a linear negative correlation with the volume of the enucleated prostate tissue. Our data also revealed that the patients in the low-BI group had the highest enucleated tissue percentage, significantly higher than that of the medium- and high-BI groups, which indicates that effective bleeding control facilitates the enucleation of a larger proportion of the prostate adenoma. 

Studies have reported that the infection rate after endoscopic surgery for BPH can reach 15% [40]. Risk factors for UTI include aging, catheterization, complicated diabetes, and long-term indwelling catheter after the operation [40]. Notably, our study is the first to identify the association between the severity of surgical bleeding and postoperative UTI. Our study revealed a link between reduced postoperative bleeding and the reduced incidence of UTI 1 month after operation. This may be attributable to the repeated and profound coagulation caused by thermal energy that is required to stop massive hemorrhagic bleeding during the procedure. Prostate bleeding cannot be controlled through suturing during an endoscopic procedure. Thermal damage or tissue necrosis caused by repeated coagulation hemostasis to the prostate tissue is likely to be the leading cause of UTI.

After the operation, patients’ voiding quality continued to improve with time; the improvement in IPSS at 3 and 6 months postoperatively was greater than that at 2 weeks postoperatively, and the improvement in VV at 6 months postoperatively was greater than that at 2 weeks and 3 months postoperatively. The improvement in PVR at 6 months postoperatively was greater than that at 2 weeks postoperatively. Although more severe bleeding during the operation was linked to less improvement in Qmax, the difference did not reach statistical significance. Contrarily, the improvement in IPSS, VV, and PVR had no obvious correlation with the severity of surgical bleeding. We speculated that effective bleeding control may assist in the enucleation of a larger proportion of the prostate adenoma, which may lead to larger Qmax improvements. Our findings are in line with those of Geavlete et al., who also reported that the removal of a larger proportion of tissue in BPH surgery can lead to a greater improvement in Qmax postoperatively, but such improvement was not observed in relation to IPSS, VV, and PVR [41]. 

This study has some limitations because of the research design. First, the patient numbers enrolled in the study were not sufficiently large to determine whether comorbidity factors before surgery were related to the severity of surgical bleeding. In addition, the follow-up time was only 6 months, and longer follow-up is necessary for functional outcome evaluation. Second, whether treatment with 5α-reductase inhibitor can affect the severity of surgical bleeding was not investigated in our study. This topic has been discussed in the literature, but no definite consensus has been reached [42,43,44]. However, our study is both novel and practical and offers a simple and valid method for determining the severity of bleeding during endoscopic prostate surgery. We identified variables that could impact bleeding severity and how the intensity of surgical bleeding influences surgical outcomes for patients. Our findings serve as a valuable reference for clinicians.

## 5. Conclusions

We can objectively determine the extent of blood loss during prostate enucleation surgery by measuring and calculating the BI. Our research revealed that patients who undergo surgery performed using a thulium laser, who are aged older than 80 years, and who have a preoperative Qmax of more than 10 cc/s exhibit relatively mild surgical bleeding. Patients’ treatment outcomes may differ depending on the severity of surgical bleeding. When patients experience less blood loss, a larger proportion of prostate tissue can be surgically removed, and the patients have a lower risk of UTIs and a larger improvement in Qmax.

## Figures and Tables

**Figure 1 diagnostics-13-00592-f001:**
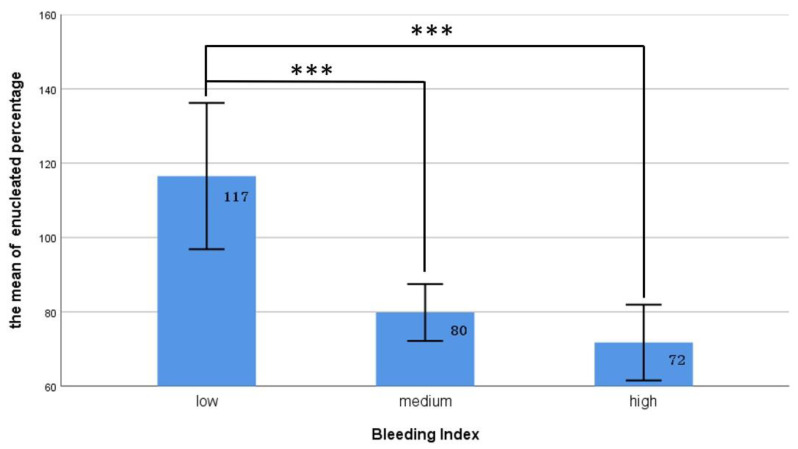
The relationship between different BI groups and the proportion of enucleated prostate tissue. The ratio of enucleated prostate tissue in the low BI group was significantly lower than that in the other two groups. *** represents *p* < 0.001.

**Figure 2 diagnostics-13-00592-f002:**
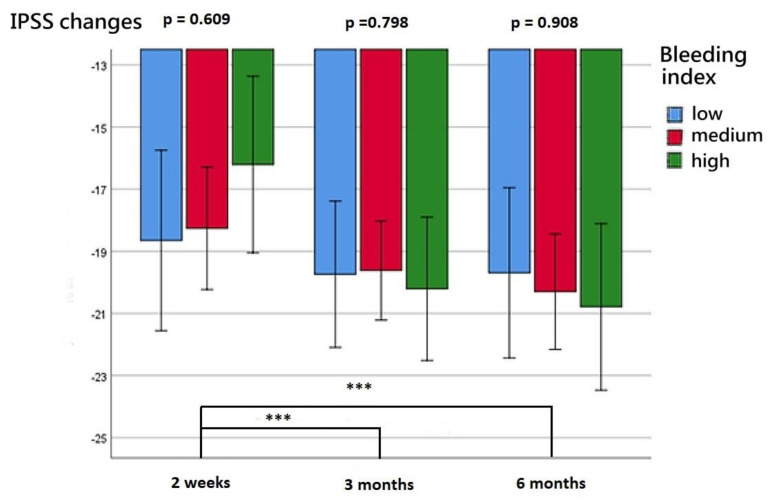
Changes in IPSS at postoperative week 2, month 3, and month 6 for the three BI groups. *** represents *p* < 0.001.

**Figure 3 diagnostics-13-00592-f003:**
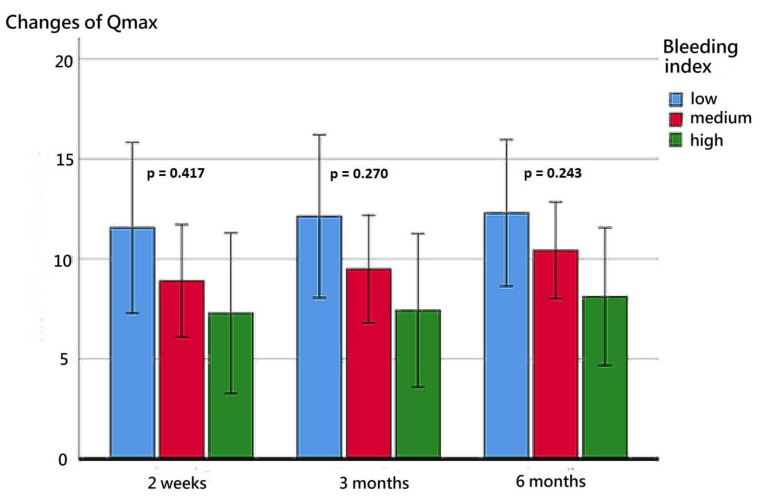
Changes in the Qmax at postoperative week 2, month 3, and month 6 for the three BI groups.

**Figure 4 diagnostics-13-00592-f004:**
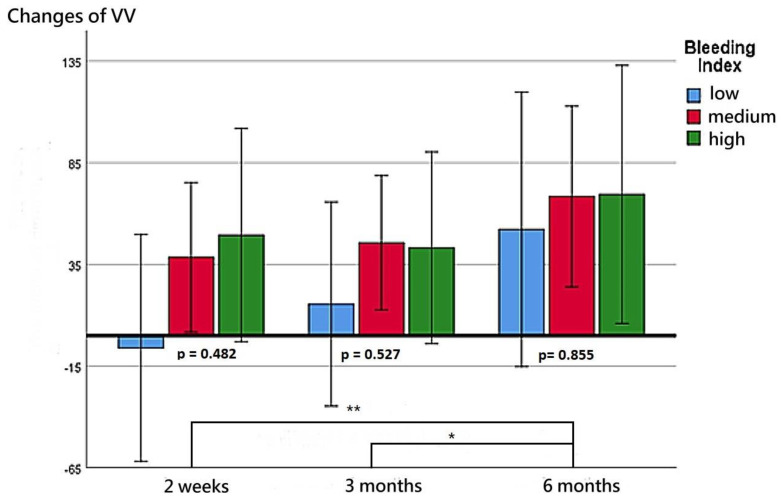
Changes in VV at postoperative week 2, month 3, and month 6 for the three BI groups. ** represents *p* < 0.01. * represents *p* < 0.05.

**Figure 5 diagnostics-13-00592-f005:**
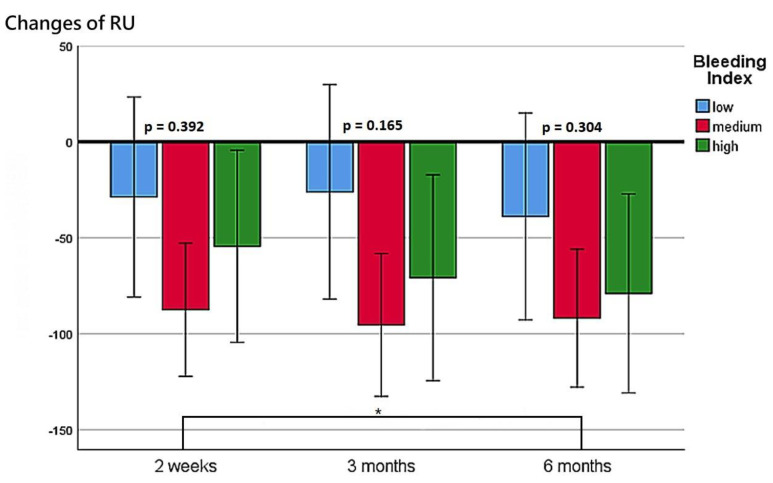
Changes in PVR at postoperative week 2, month 3, and month 6 for the three BI groups. * represents *p* < 0.05.

**Table 1 diagnostics-13-00592-t001:** Baseline data of the patients (continuous variables) and correlations with the BI.

Variables	Mean ± SD	P.C.	*p*-Value
**Pre-op**			
Age (years)	67.7 ± 8.7	−0.127	0.072
PSA (ng/mL)	5.9 ± 6.13	0.029	0.685
Cr (mg/dL)	1.02 ± 0.57	0.132	0.061
Prostate volume, total (mL)	53.2 ± 21.2	0.042	0.552
Prostate volume, T zone (mL)	24.7 ± 14.7	0.009	0.897
IPSS score, total	24.4 ± 4.7	−0.051	0.471
IPSS score, voiding	15.2 ± 3.3	−0.025	0.772
IPSS score, storage	9.2 ± 4.3	−0.046	0.521
Qmax (mL/s)	8.4 ± 3.8	−0.081	0.275
VV (mL)	193.4 ± 108.0	−0.004	0.953
PVR (mL)	112.4 ± 151.0	0.006	0.932
Medication duration (months)	27.7 ± 40.6	−0.022	0.760
**Peri-op**			
Tissue enucleated percentage (%)	86.5 ± 46.7	−0.253	<0.001 *
Operation time (min)	84.9 ± 40.6	−0.008	0.910
Post-op hospital stays (days)	2.2 ± 0.6	0.053	0.455

Abbreviations: SD: standard deviation; P.C.: Pearson correlation; PSA: prostate-specific antigen; Cr: creatinine; T: transitional; IPSS: International Prostate Symptom Score; Qmax: maximum urinary flow rate; VV: viding volume; PVR: post-voiding residual urine; *: significant different.

**Table 2 diagnostics-13-00592-t002:** Baseline data of the patients (categorical variables) and the corresponding BI.

Characteristics		Bleeding Index		T	*p*-Value
		Mean	SD		
**Thulium laser usage**	No	11.20	11.39	4.36	<0.001 *
	Yes	5.73	6.27		
**Admitted with catheter**	No	8.41	9.77	1.16	0.248
	Yes	6.35	5.33		
**U.R in the past 3 months**	No	8.35	9.79	0.56	0.577
	Yes	7.58	8.07		
**Age (years)**	<80	8.36	9.60	2.432	0.019 *
	≥80	5.59	3.98		
**Prostate volume (mL)**	<80	7.71	8.74	−1.43	0.152
	≥80	10.36	11.63		
**Qmax**	<10	9.08	10.51	2.68	0.008 *
	≥10	6.08	5.40		
**Comorbidities**					
**DM**	No	7.67	9.07	−1.25	0.212
	Yes	9.67	9.74		
**HTN**	No	7.67	9.86	−0.67	0.505
	yes	8.54	8.51		
**CAD**	No	7.94	9.31	−0.73	0.466
	yes	9.65	8.42		
**CHF**	No	8.11	9.32	0.25	0.802
	yes	7.06	4.71		
**Arrythmia**	No	8.00	9.33	−0.47	0.641
	yes	9.20	8.00		
**Stroke**	No	7.76	9.08	−1.67	0.096
	yes	11.65	10.31		
**CRI**	No	8.02	9.28	−0.39	0.695
	yes	9.06	8.75		

Abbreviations: SD: standard deviation; U.R: urinary retention; Qmax: maximum urinary flow rate; QoL: International Prostate Symptom Score-QoL index. DM: diabetes mellitus; HTN: essential hypertension; CAD: coronary artery disease; CHF: congestive heart failure; CRI: chronic renal insufficiency; *: significant different.

**Table 3 diagnostics-13-00592-t003:** The relationship between the three BIs and surgical outcomes.

Outcomes		Bleeding Index			X^2^	*p*-Value
		Low	Medium	High		
**Days of post-op hospital stays**	≤2 days	43(89.6%)	91(88.4%)	37(72.6%)	7.765	0.079
	3 days	3(6.3%)	9(8.7%)	11(21.6%)		
	≥4 days	2(4.1%)	3(2.9%)	3(5.8%)		
**Urethral stricture**	No	46(95.8%)	96(93.2%)	48(94.1%)	0.38	0.925
	Yes	2(4.2%)	7(6.8%)	3(5.9%)		
**UTI within 1 months**	No	39(81.2%)	63(61.2%)	35(68.6%)	6.07	0.046 *
	Yes	9(18.8%)	40(38.8%)	16(31.4%)		
**UR within 1 months**	No	46(95.8%)	95(92.2%)	46(90.2%)	1.134	0.287
	Yes	2(4.2%)	8(7.8%)	5(9.8%)		
**Return to ER within 1 month**	No	42(87.5%)	89(86.4%)	45(88.2%)	0.109	0.962
	Yes	6(12.5%)	14(13.6%)	6(11.78%)		

Abbreviations: BI: bleeding index; UTI: urinary tract infection; UR: urinary retention; ER: emergency room; *: significant different.

## Data Availability

The data used to support the findings of this study are available from the corresponding author upon request.

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
