# Peer review of "Simple and Convenient Method for Assessing the Severity of Bleeding during Endoscopic Prostate Surgery and the Relationships between Its Corresponding Surgical Outcomes"

_diagnostics, 2023, doi:10.3390/diagnostics13040592_

Round 1
Reviewer 1 Report
The study is consistent and may find a place in clinical practice.
Several studies from the same authors (Hou and Chen) are cited. The authors should retain only the relevant cited studies in the references of the present manuscript.
Author Response
Dear Reviewers:
We agree with you that the references of Hou CP and Chen JW were cited twice by us, respectively, so we have deleted the following paper citations in our Reference. The serial numbers of all references have been corrected accordingly.
Hou CP, Lin YH, Yang PS, et al. Clinical Outcome of Endoscopic Enucleation of the Prostate Compared with Robotic-Assisted Simple Prostatectomy for Prostates Larger Than 80 cm3 in Aging Male. Am J Mens Health. 2021;15(6):15579883211064128
Chen JW, Lin WJ, Lin CY, Hung CL, Hou CP, Cho CC, Tang CY. Automated classification of blood loss from transurethral resection of the prostate surgery videos using deep learning technique. Applied Sciences, 2020, 10.14: 4908.
Thank you for your valuable suggestions. We look forward to your reply and the agreement to have our manuscript considered for acceptance.
Sincerely
Chen Pang Hou
Reviewer 2 Report
Interesting concept and analysis. Any reason you didn't include holmium enucleation and only thulium? Surgeon preference?
Interesting conclusions and correlations.
Author Response
Answer to reviewer 2:
Dear Reviewers:
Thank you very much for your advice. As far as we know, Holmiun laser prostate enucleation is a pervasive operation in European countries, and there are many related literature reports. However, in the country where I practice -Taiwan, there are very few cases of prostate enucleation using the Holmiun laser. Many large medical centers even do not purchase a Holmiun Laser. On the contrary, we have used a Thulium laser to perform prostate enucleation in many cases, so we researched Thulium laser prostate enucleation.
Thank you for your valuable suggestions. We look forward to your reply and the agreement to have our manuscript considered for acceptance.
Sincerely
Chen Pang Hou